# Difference in Movement between Superficial and Deep Parts of the Infrapatellar Fat Pad during Knee Extension

**DOI:** 10.3390/jfmk6030068

**Published:** 2021-08-14

**Authors:** Syoya Nakanishi, Ryosuke Morimoto, Masashi Kitano, Kengo Kawanishi, Arisa Tanaka, Shintarou Kudo

**Affiliations:** 1Department of Physical Therapy, MORINOMIYA University of Medical Sciences, Osaka 559-8611, Japan; n.syoya.pt@gmail.com (S.N.); patya-patyaswim-tigers@outlook.jp (R.M.); 2Graduate School of Health Sciences, MORINOMIYA University of Medical Sciences, Osaka 559-8611, Japan; kitakita1215reha@yahoo.co.jp (M.K.); riverwest1004@yahoo.co.jp (K.K.); 3AR-Ex Oyamadai Orthopedic Clinic Tokyo Arthroscopy Center, Tokyo 158-0082, Japan; arinco.7522@gmail.com; 4Inclusive Medical Science Research Institute, MORINOMIYA University of Medical Sciences, Osaka 559-8611, Japan; 5AR-Ex Medical Research Center, Tokyo 158-0082, Japan

**Keywords:** infrapatellar fat pat, movement, superficial part, deep part, flow velocity, knee

## Abstract

(1): The superficial and deep parts of the infrapatellar fat pat (IFP) have different morphological and functional characteristics. Knee pain often occurs during movement, and it is important to clarify the movement of the IFP during knee joint movement. The purpose of this study is to clarify that the movement of the superficial and deep parts of the IFP are different during knee extension in vivo using ultrasonography (US). (2): US was performed on 15 knees of 15 healthy adults. The probe was placed longitudinally at the center of the patellar tendon and the IFP was imaged. Measurements were taken during active extension of the knee from 90 degrees to 10 degrees of knee flexion at a rate of 30 times/min. The captured US videos were analyzed using Flow particle image velocimetry (Flow PIV) fluid measurement software. The region-of-interest (ROI) was set at the superficial part and the deep part of the IFP, and the flow velocity was calculated for each. (3): The flow velocity of the deep part (1.37 ± 0.13 cm/s) of the IFP was significantly faster than that of the superficial part (0.80 ± 0.23 cm/s). (4): Our results show that the flow velocity of the IFP is different between the superficial and deep parts and that US may be a better assessment tool for the movement of the IFP.

## 1. Introduction

The infrapatellar fat pat (IFP) of the knee is intracapsular, but extra-synovial. The IFP occupies the anterior part of the knee and exists between the patellar tendon, the femoral condyles, and the tibial plateau [1,2]. The posterior genicular nerve and the posterior tibial nerve pass the external part of the meniscus, the synovium, and the cruciate ligaments, and innervate the IFP [3]. The IFP functions in regulating the contact pressure of the knee, providing neurovascular supply to adjacent structures, and there is a buffering effect due to morphological changes [2]. Therefore, the IFP is known to be a factor in severe knee pain. The IFP has been reported to be fibrotic in anterior knee pain syndrome with both osteoarthritis (OA) [4] and anterior cruciate ligament (ACL) reconstruction [5,6,7]. In other words, after knee OA or ACL reconstruction, morphological changes of the IFP are inhibited during knee motion, which may increase the internal pressure of the IFP and cause pain due to impingement [8]. Thus, understanding the morphological changes in the IFP during knee motion is important to treat anterior knee pain after knee OA and ACL reconstruction.

The superficial and deep parts of the IFP have different morphological and functional characteristics [7,9], and the functional and morphological characteristics of both need to be evaluated separately when considering the problems that arise in the IFP.

Several reports have used MRI and 3D models to evaluate the IFP [10,11]. While these methods can be used to confirm accurate information, they are time-consuming to evaluate. In recent years, some studies have confirmed morphological changes in the superficial and deep part of IFP using ultrasound (US), which is easier to perform and requires less time for evaluation [7,8,12,13]. However, these studies measured the morphology of the IFP before and after changing the flexion angle of the knee joint. In other words, the morphology of the IFP due to differences in joint angle can be observed, but the results are static rather than dynamic. In actual clinical situations, pain often occurs during movement, and it is important to clarify movements of the IFP during knee extension movement.

We hypothesized that because the morphological and functional characteristics of the superficial and deep parts of the IFP differ, there may be differences in movement between the two regions. The purpose of this study was to clarify the differences in movement between the superficial and deep parts of the IFP during knee extension in vivo using US.

## 2. Methods

A total of 15 knees of 15 healthy normal volunteers participated in this study. Subjects with history of orthopedic or neurological disorders were excluded. The characteristics of the subjects are summarized in Table 1. Informed consent was obtained from all participants in the study. The study was approved by the Ethics Committee of our institution (authorization number: 2019-035).

### 2.1. Procedures

Imaging of the IFP was assessed by ultrasonography (US) (Aplio300, Canon medical systems Co, Tokyo, Japan) with a 14 MHz linear probe (PLT-1005BT). The gain, dynamic range, and frame rate were the same throughout all measurements (gain: 100, dynamic range: 70, frame rate: 30 flame per minutes). The probe was placed longitudinally at the center of the patellar tendon in the sitting position (Figure 1 and Figure 2). The probe was fixed manually to capture the dynamic image with identification of the high echo line of the patella during knee extension. A stick marker was set in front of the knee to control the extension movement of the knee at 10 degrees of flexion using a goniometer. For the movement task, the knee was actively extended from 90 degrees to 10 degrees of knee flexion and back to 90 degrees of flexion at a rate of 30 times/min, and measurements were collected twice. We confirmed that the subjects were able to maintain a constant rhythm after practice.

Captured US videos were analyzed using Flow particle image velocimetry (Flow PIV) fluid measurement software (Library, Tokyo, Japan) with PIV setting of tracking pixels 15 × 21 at two frame intervals). A distinction was made between superficial and deep parts based on Kitagawa et al. [8] report. The superficial part of the IFP was a low-echo intensity area above the high-intensity septum in the middle region of the IFP. The region-of-interest (ROI) was set at both the superficial and deep parts of the IFP. The vertical width of the superficial part was 30 pixels deep from the deep edge of the patellar tendon, and the lateral width was the space starting from 75 pixels distal to the inferior patellar pole and extending a further 150 pixels distally. The vertical width of the deep part was 30 pixels deep from the high-intensity septum in the middle region of the IFP, and the lateral width was similar to the superficial part (Figure 2). The flow velocity of each part of the IFP was arranged by time sequence data (Figure 3). The average peak velocity of the IFP was determined from two knee extension movements performed in the superficial and deep parts.

### 2.2. Reliability

The reliability of assessing movements of the IFP was examined. The measurements were performed by two physical therapists. Both examiners had 2 years’ experience in US. Before conducting this study, they practiced imaging the IFP during knee extension for about 1 h in one healthy male subject, who was not included in the study. The IFP was imaged so that the linear high-echo of the patellar pole and tibial tuberosity on the US image would not shift during knee extension. The test was then conducted on a different day. Two examiners measured each healthy subject (*n* = 15) twice according to the above method. One of the subjects was measured twice the next day using the same methods. Furthermore, the two examiners performed the measurements so that they did not know each other’s measurements.

### 2.3. Statistical Analysis

The reliability of the measurement method was calculated by the intra-rater correlation coefficient (ICC) (1.2) and the inter-rater correlation coefficient (ICC)(2.2) and was assessed as described previously (Fleiss, 1981). Measurement error was calculated using the Standard Error of the Measurement (SEM) and MDC95, which is the 95% confidence interval of Minimal Detectable Change (MDC). SEM was calculated using the following formula: SEM = the standard deviation × $\sqrt{(1-\mathrm{ICC})}\;$ [14]. MDC was obtained employing the following equation: MDC = 1.96 × $\sqrt{2}$ × SEM [14]. To compare the flow velocities of the superficial and deep parts of the IFP between the two groups, the Shapiro–Wilk test was performed, and, if the normal distribution was confirmed, the paired *t*-test was performed. When the normal distribution was not confirmed, the Wilcoxon test was performed. The significance level was set at *p* < 0.05. SPSS Statistics ver.25.0 (IBM. Corp, Tokyo, Japan) was used for all statistical analyses.

## 3. Results

Results of the intra-rater correlation coefficient, the inter-rater correlation coefficient, SEM, and MDC95 are shown in Table 2. Since the Shapiro–Wilk test did not show normal distribution, the Wilcoxon test was applied. As a result, the flow velocity of the superficial part of the IFP in the control group was 0.80 ± 0.23 cm/s and that of the deep part was 1.37 ± 0.13 cm/s (Table 3). Both the superficial and deep parts of the IFP move infero-superficially during knee extension and the flow velocity of the deep part was significantly faster than that in the superficial part.

## 4. Discussion

The purpose of this study was to clarify differences in movement between the superficial and deep parts of the IFP during knee extension in vivo using ultrasonography. As a result, the flow velocity of the IFP was significantly faster in the deep part than that in the superficial part. Recently, there have been several studies examining the morphological changes of the IFP at different knee positions. Mikklineni et al. [13] measured the compressibility of the largest adipose lobule by the changes in anterior-posterior length at 0 and 90 degrees of knee flexion. Macchi et al. [7] assessed the thickness of adipose lobules of the IFP divided into superficial (macro-chambers) and deep (micro-chambers) parts, respectively, at 0 and 90 degrees of knee flexion. Kitagawa et al. [8] also divided the IFP into superficial and deep parts and measured the difference in these structures between 10 and 90 degrees of knee flexion, respectively. In other words, these studies clarified the morphological changes of the IFP at different knee position. However, movement of the IFP during knee extension in vivo remained unclear. Our results showed that both the superficial and deep parts of the IFP move infero-superficially during knee extension, and the flow velocity of the deep part of the IFP is faster than that of the superficial part during knee extension.

Flow PIV fluid measurement software enables fluid measurement from US image data and can determine the relative movement between consecutive adjacent frames of US images, visualize the flow, and measure the velocity vector to quantify the dynamic changes. In recent years, PIV has been used to quantify the proximal and distal gliding properties of the median nerve and to determine decreased gliding properties of the vastus lateralis and subcutaneous tissue in patients with residual pain after trochanteric fracture surgery [15,16]. Measurements in this study had excellent reliability for both the intra-rater correlation coefficient and the inter-rater correlation coefficient. By using PIV, we were able to measure and quantify the movement of the superficial and deep parts of the IFP during active knee joint extension.

Morphological changes of the IFP are known to differ between the superficial part (macro-chamber) and the deep part (micro-chamber) [7]. The macro-chamber at the superficial part of the IFP, which is increased by the forward rolling of femur condyle, can be deformed to buffer the pressure at the anterior interval during knee extension. However, the micro-chamber has many collagen fibers and is less likely to be deformed by changes in pressure. Studies using 3D models have reported that the IFP is extruded into the condyle of the femur and moves anteriorly and inferiorly during knee extension [11]. Therefore, it is thought that the micro-chamber, which has little morphological change, glides forward and changes the pressure exerted by the condyle of the femur during knee extension. In other words, the micro-chamber requires greater gliding than the macro-chamber.

This study is the first to establish a method to quantitatively measure the movements of the superficial and deep parts of the IFP during knee extension in healthy subjects. Our results indicate that movement of the deep part of the IFP had larger than that of the superficial part. These results suggest that, when assessing the IFP during knee movement, it is necessary to separate the IFP into superficial and deep parts, rather than assessing it as a single structure. The results of this study may help to assess the movement of the IFP in knee OA and after ACL reconstruction.

### Limitations

This study had some limitations. First, measurement with a fixation device is difficult to perform, because the patella moves proximally during knee extension. Second, in daily life, weight-bearing such as walking, climbing stairs, and squats affect the movement of the IFP, and, in this study, US images of the IFP were obtained during non-weight-bearing. Further studies are needed to evaluate the movement of the IFP during knee exercise under weight-bearing conditions.

## 5. Conclusions

Movement of the superficial and deep parts of the IFP during knee extension was measured quantitatively using this method. We observed that the flow velocity of the IFP is different between the superficial and deep parts and that the flow velocity in the deep part was faster than that in the superficial part. Morphological differences between the superficial and deep parts of the IFP may affect flow velocity. In this study, we were able to clarify the movements of the IFP during knee motion in healthy subjects, which may help to clarify the pathogenesis of anterior knee pain.

## Figures and Tables

**Figure 1 jfmk-06-00068-f001:**
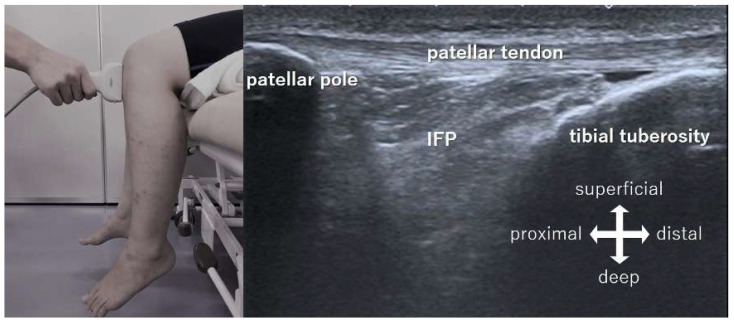
Ultrasound image capturing method. The probe was placed longitudinally at the center of the patellar tendon in the sitting position. IFP: infrapatellar fat pad.

**Figure 2 jfmk-06-00068-f002:**
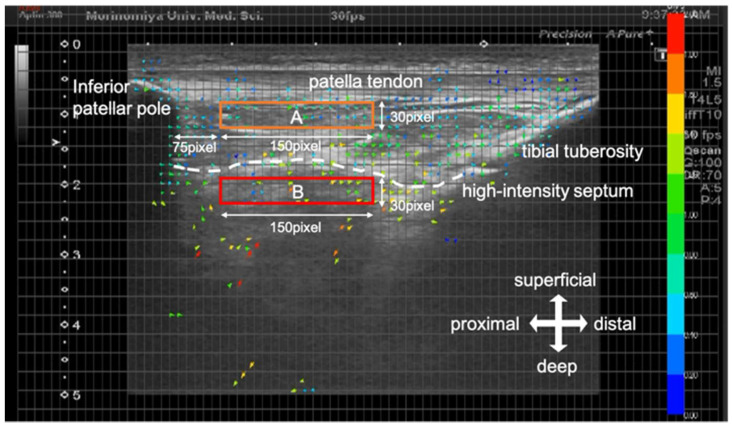
Ultrasound image using flow particle image velocimetry. Based on an ultrasonogram movie image, 2 regions of interest were placed on the IFP superficial part and IFP deep part. (**A**) superficial part. (**B**) deep part.

**Figure 3 jfmk-06-00068-f003:**
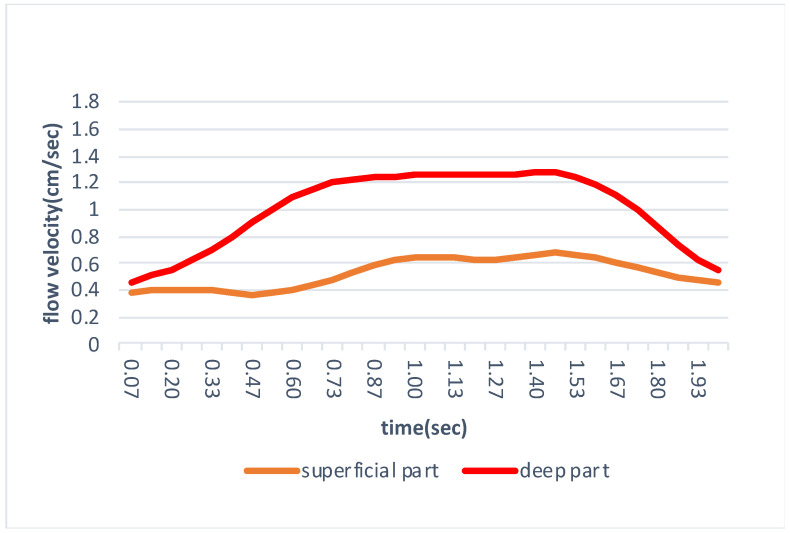
Time series data of the IFP superficial part and the IFP deep part.

**Table 1 jfmk-06-00068-t001:** The characteristics of the subjects.

Variable	
Sex (male:female)	8:7
Age (years)	20.6 ± 0.5
Height (cm)	164.5 ± 10.4
Weight (kg)	60.4 ± 14.0
Body mass index (kg/m²)	22.1 ± 3.5

Data indicate mean ± standard deviation.

**Table 2 jfmk-06-00068-t002:** Intra-rater correlation coefficient and the inter-rater correlation coefficient.

**Intra-Rater Correlation Coefficient**	**ICC(1.2)**	**95%CI**	**df**	**SEM(m/s)**	**MDC_95_(m/s)**
flow velocity of superficial part	0.92	0.78–0.97	14.00	0.06	0.16
flow velocity of deep part	0.82	0.56–0.94	14.00	0.08	0.21
**Inter-Rater Correlation Coefficient**	**ICC(2.2)**	**95%CI**	**df**	**SEM(m/s)**	**MDC_95_(m/s)**
flow velocity of superficial part	0.96	0.89–0.98	14.00	0.06	0.16
flow velocity of deep part	0.89	0.66–0.96	14.00	0.05	0.15

ICC(1.2): Intra-rater correlation coefficient. ICC(2.2): inter-rater correlation coefficient. SEM: Standard Error of the Measurement. MDC: Minimal Detectable Change.

**Table 3 jfmk-06-00068-t003:** Data of flow velocity.

	Superficial Part	Deep Part	*p*-Value
flow velocity(cm/s)	0.80 ± 0.21	1.33 ± 0.10	<0.05

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
