# Peer review of "Difference in Movement between Superficial and Deep Parts of the Infrapatellar Fat Pad during Knee Extension"

_jfmk, 2021, doi:10.3390/jfmk6030068_

Round 1

Reviewer 1 Report

In the manuscript, the authors use ultrasonography for evaluating the difference in movement between superficial and deep parts of the infrapatellar fat pad during knee extension. Fifteen healthy volunteers were recruited in this study, and results found that the flow velocity of the deep part of the IFP was significantly faster than that of the superficial part. Overall, the study methods were reasonable and findings were important. However, some detailed experimental procedures should be described more clearly. The following are my comments:

1) How to keep the probe of ultrasonography at the same position during knee extension? It seems not a reliable way to hold the probe manually for keeping in the same position during motion. Please comment.

2) Why do not use ultrasonography to measure the thickness of the infrapatellar fat pad at the same time? The combination of the information of thickness and the flow velocity should obtain more meaningful findings.

3) The values of the gain, dynamic range, and frame rate should be provided in the manuscript.

4) The title of the figure should appear under the figure. Please correct these errors.

5) Where is Figure 2? The title of the figure is missing. Please correct it.

6) In table2, what are ICC(1.2) and ICC(2.2)?

7) The full names of abbreviations should appear for the first time shown in the manuscript, such as ICC and PIV. Please correct it.

8) In discussion, the citation format is inconsistent, like “Mikklineni et al. (2018)” and “Macchi et al.[7]”.

9) Please explain what academic values it adds to the field? Does it provide useful information for helping the relief of infrapatellar fat pad syndrome?

10) In the conclusion, the practical application field of the research findings can be described that highlights the contribution of this article.

Author Response

Response to reviewer 1:

We wish to thank the reviewer for this comment. We have answered each of your points below.

1) How to keep the probe of ultrasonography at the same position during knee extension? It seems not a reliable way to hold the probe manually for keeping in the same position during motion. Please comment.

1) The following statement has been added as a way to keep the ultrasound probe in the same position during knee extension.

The IFP was imaged so that the linear high-echo of the patellar pole and tibial tuberosity on the US image would not shift during knee extension. (Add p. 4, lines 121-123)

Using the method described above, the measurements were highly reliable. (p. 5, lines 142-143). A more reliable method is a fixation device, but as mentioned in the limitations, it cannot be used in this study (p. 6, lines 199-200).

2) Why do not use ultrasonography to measure the thickness of the infrapatellar fat pad at the same time? The combination of the information of thickness and the flow velocity should obtain more meaningful findings.

2) Thickness of the superficial part of the infrapatellar fat pad could not be measured during knee movements, because it is uncleared the septa between superficial and deep part of the infrapatellar fat pad. And we assessed the thickness of the infrapatellar fat pad at the extension and 90-degree knee flexed position. However, there are not associated between thickness and flow velocity of the infrapatellar fat pad. We hypothesized the subjects in this study were asymptomatic subjects who have normal range of both flow velocity and thickness change. We are researching the associated between those in both normal and patients with anterior knee pain in the next study.

 Therefore, your advice is represented in next study, thank you.

3) The values of the gain, dynamic range, and frame rate should be provided in the manuscript.

3) In accordance with the reviewer's comment, we have changed this to (gain:100, dynamic range:70, frame rate:30 flame per minutes (Add p. 2, lines 81-82).

4) The title of the figure should appear under the figure. Please correct these errors.

4) In accordance with the reviewer's comment, we have changed (Del p. 3-4, lines 104, 108-109, 113) (Add p. 3-4 lines 106-107, 111-112, 115)

5) Where is Figure 2? The title of the figure is missing. Please correct it.

5) In accordance with the reviewer's comment, we have changed this to Figure 1.2 has been reduced to one. Therefore, change Figure3 to Figure2, and Figure4 to Figure3 (p. 3-4, lines 106-107, 111-112, 115).

6) In table2, what are ICC(1.2) and ICC(2.2)?

Added an explanation to table2 (Add p. 5 line 150).

 ICC(1.2)=Intra-rater correlation coefficient.

ICC(2.2)=inter-rater correlation coefficient.

7) The full names of abbreviations should appear for the first time shown in the manuscript, such as ICC and PIV. Please correct it.

7) In accordance with the reviewer's comment, we have changed (p. 1,3,5, lines 22-23, 91-92, 130).

8) In discussion, the citation format is inconsistent, like “Mikklineni et al. (2018)” and “Macchi et al.[7]”.

8) In accordance with the reviewer's comment, we have changed (p. 5, line 158).

9) Please explain what academic values it adds to the field? Does it provide useful information for helping the relief of infrapatellar fat pad syndrome?

9) Because the answer overlaps with 10), I entered my answer in 10).

10) In the conclusion, the practical application field of the research findings can be described that highlights the contribution of this article.

10) In accordance with the reviewer's comment, we have changed conclusion (Add p. 6-7, line 210-212).

In this study, we were able to clarify the movements of the IFP during knee motion in healthy subjects, which may help to clarify the pathophysiology of anterior knee pain.

We hope the revised version is now suitable for publication and look forward to hearing from you in due course.

Sincerely,

Shintarou Kudo

Reviewer 2 Report

The purpose of this study is to clarify that the movement of the superficial and deep parts of the IFP are different during knee extension in vivo using ultrasonography (US). US was performed on 15 knees of 15 healthy  adults. Measurements were taken during active extension of the knee from 90 degrees to 10 degrees of knee flexion at a rate of 30 times/min. The captured US videos were analyzed using Flow PIV fluid measurement software. The results show that the flow velocity of the IFP is different between the superficial and deep parts and that US may be a better assessment tool for the movement of the IFP.

The study is interesting. I would like to recommend its publication. I have several minor comments:

  1. The association between IFP and knee pain should be simply addressed in the introduction.
  2. Why did the authors only choose one knee from one participant? Please give the reason.
  3. The setting of the ultrasound machine should be provided.
  4. In Figure 3, please specify whether that is the superior or inferior patellar pole.
  5. Please detail how to differentiate the superficial and deep intra-patellar fat pad on ultrasound. Is there any visual guide?
  6. Please provide the formula regarding how to calculate MDC and SEM. The author can reference the following article also evaluating MDC and SEM: Ultrasound measurements of superficial and deep masticatory muscles in various postures: reliability and influencers. Sci Rep 2020

Author Response

Response to reviewer 2:

We wish to thank the reviewer for this comment. We have answered each of your points below.

1) The association between IFP and knee pain should be simply addressed in the introduction.

1) In accordance with the reviewer's comment, we have changed (Add p. 2, lines 45-47).

Before correction

In osteoarthritis (OA) of the knee, the IFP causes fibrosis in the synovium and may be related to the pathophysiology of OA[4]. The IFP in knee OA can be compressed by narrowing of the joint space, which may increase the internal pressure of the IFP. In addition, anterior cruciate ligament (ACL) reconstruction may create fibrosis in the IFP by prolonging inflammation at the arthroscope insertion site, which may create adhesions in the patellar tendon, the patellar retinaculum, the IFP, the anterior synovium, and the bursa[5][6][7]. (Del p. 1-2, lines 39-45)

After correction

The IFP has been reported to be fibrotic in anterior knee pain syndrome with both osteoarthritis (OA) [4] and anterior cruciate ligament (ACL) reconstruction[5–7] (Add p. 2, lines 45-47).

2) Why did the authors only choose one knee from one participant? Please give the reason.

2) If both sides are measured, there is a possibility of bias if there are samples with the same height, weight, age, etc. In the lower limb, the function of unilateral limb had influenced on another limb. Because of this, we decided to make it one person, one knee.

3) The setting of the ultrasound machine should be provided.

3) In accordance with the reviewer's comment, we have changed this to (gain:100, dynamic range:70, frame rate:30 flame per minutes) (Add p. 2, lines 81-82).

4) In Figure 3, please specify whether that is the superior or inferior patellar pole.

4) In accordance with the reviewer's comment, we have changed (p. 3-4, lines 98, 110).

5) Please detail how to differentiate the superficial and deep intra-patellar fat pad on ultrasound. Is there any visual guide?

5) A distinction was made between superficial and deep parts based on Kitagawa et al. report. the superficial part of the IFP was a low-echo intensity area above the high-intensity septum in the middle region of the IFP (Add p. 3, lines 93-95).

 Kitagawa, T.; Nakase, J.; Takata, Y.; Shimozaki, K.; Asai, K.; Tsuchiya, H. Use of Ultrasonography to Evaluate the Dynamics of the Infrapatellar Fat Pad after Anterior Cruciate Ligament Reconstruction: A Feasibility Study. Journal of Medical Ultrasonics 201946, 147–151, doi:10.1007/s10396-018-0917-7. 

6) Please provide the formula regarding how to calculate MDC and SEM. The author can reference the following article also evaluating MDC and SEM: Ultrasound measurements of superficial and deep masticatory muscles in various postures: reliability and influencers. Sci Rep 2020

6) SEM was calculated using the following formula: SEM = the standard deviation ×√(1 − ICC). 

MDC was obtained employing the following equation: MDC=1.96×√2×SEM (Add p. 5, lines 128-130).

We hope the revised version is now suitable for publication and look forward to hearing from you in due course.

Sincerely,

Shintarou Kudo

Round 2

Reviewer 1 Report

The authors corrected the manuscript in the latest version and answered all questions well. After the revision, the quality of the manuscript was also imrpoved.